# Integrative Approaches Establish Colour Polymorphism in the Bamboo-Feeding Leafhopper *Mukaria splendida* Distant (Hemiptera: Cicadellidae) from India

**DOI:** 10.3390/insects14010044

**Published:** 2023-01-02

**Authors:** Mogili Ramaiah, Naresh M. Meshram, Debjani Dey

**Affiliations:** 1Department of Plant Protection, ICAR-Krishi Vigyan Kendra, Acharya N. G. Ranga Agricultural University, Kalyandurg 515761, India; 2Division of Entomology, ICAR-Indian Agricultural Research Institute, New Delhi 110012, India; 3Department of Entomology, ICAR-Central Citrus Research Institute, Nagpur 440033, India

**Keywords:** *Mukaria splendida*, bamboo, colour polymorphism, distribution, phytoplasmal vector, India

## Abstract

**Simple Summary:**

Authentic species-level identification is key to many fields of entomology, especially insect pest management and biosecurity. Bamboo, known as “Poor Man’s Timber” and “Green Gold”, is one of the fastest growing plants on earth. Bamboo and food grain crops (rice and wheat, for example) belong to the same family (Poaceae), so there is a chance of host shift because of global climate change and extensive cultivation. In the present study, the colour polymorphism of the bamboo-feeding leafhopper *Mukaria splendida* is highlighted, along with additional geographical distribution records. The integration of the morpho-molecular data confirmed that all the colour morphs belonged to the same species, *M. splendida*.

**Abstract:**

The leafhopper species, *Mukaria splendida* Distant, is economically important due to itsstatus as a pest on bamboo and was recently reported to bea vector for phytoplasmal disease. Morphological identification is often difficult and requires a high level of taxonomic expertise, with misidentifications causing problematic false-positive/negative results. In this paper, colour polymorphism was recorded in the bamboo-feedingleafhopper *M. splendida* (Distant, 1908), which is a major insect pest in the bamboo ecosystem based on explorations conducted in different locations of India. Ten morphs were identified for each sex of *M. splendida* Distant based on the colour pattern on the pronotum and forewings. However, in view of the economic importance of the species, the morphological studies were integrated with the molecular data for the accurate identification of the species. The morphological characteristics and sequence results of the amplified product of the mitochondrial cytochrome oxidase subunit I (mtCOI) gene confirmed that all the morphs were *M. splendida* and the pairwise distance matrix showed a negligible genetic distance in the COI mtDNA gene. Simulated future predictions, along with detailed notes on the colour polymorphic forms with illustrations, and additional distribution records as well as thebiology of *M. splendida* were discussed in light of the available literature, all of which will aid the authentic identification of this species.

## 1. Introduction

India has a rich diversity of bamboos (Poaceaefamily), standing second globally only next to China [1]. Bamboos provide a sustainable livelihood to billions and form an integral part of the social and cultural fabric of the Indian subcontinent [2]. The seven sister states of north-eastern India are rich in bamboo diversity, with more than 40% of the total land area covered with bamboo. Nearly 90 bamboo species are found distributed in the north-eastern hilly states of India, with 41 of them being native tothe area. In India, the three major genera of bamboo (*Bambusa*, *Dendrocalamus* and *Ochlandra*), each containing more than ten species, together account for about half of all bamboo species recordedin India [3]. Manipur stands first witha bamboo area coverage of 44.35%; the state with the least bamboo area coverage is Uttar Pradesh (0.51%) (Forest Report, 2019). Insect pests such as weevils, beetles, termites and sap-sucking insects such as leafhoppers and aphids cause severe damage to the plant [4].

Leafhoppers are one of the important auchenorrhynchan groups of the order Hemiptera that breedon bamboo, but the knowledge about them is scanty in India. A perusal of all available literature on leafhoppers infesting bamboo indicated that the majorityare members of the Typhlocybinae [5,6,7,8,9], Cicadellinae [10,11], Nirvaninae [12,13], Evacanthinae [11] and Mukariinae [13,14,15] subfamilies. The Mukariinitribe, which was established by Distant (1908), was recently transferred to the Deltocephalinae subfamily, based on molecular and morphological data [16,17,18]. Based on their investigation, Yang et al. (1999) concluded that the most important leafhopper pests infesting bamboo belonged to the Nirvaninae subfamily [19]. Li and Chen (1999) described several Chinese Nirvaninae breeding on bamboo, including three genera and nine species of *Flatfronta* Chen and Li (placed in the Nirvaninii tribe as one species), *Mohunia* Distant (placed in the Mukariini tribe, with four species) and *Mukaria* Distant (placed in the Mukariini tribe, with four species) [13]. Praveen and Sabita (2019) observed a morphological variation (only four morphs) in the populations of the leafhopper *Mukaria splendida* Distant [20]. Recently, Meshram et al. (2020), Ramaiah and Meshram (2021) and Meshram (2021) added four new species (placed in the Mukariini tribe) from India [21,22,23].

Li and Chen (1999) pointed out that most members of the Mukariini tribe feed exclusively on Bambusoideae and are among the major pests of bamboo [13]. The *Mukaria* genus is characterized by a depressed body and anteriorly rounded vertex; it was established by Distant (1908) based on the *M. penthiomiodes* type of species. Zahniser and Dietrich (2013) provided a revised interpretation of the Deltocephalinae subfamily and reviewed the Mukariini tribe [16]. This group has received the attention of several workers across the world [17,24,25,26].

Bamboo and important food grain crops such asrice and wheat belong to the same family, Poaceae. The chances of host shift due to global climate change and extensive cultivation cannot be ruled out. This study helps in recording their potential pests. Out of the reported leafhoppers associated with bamboo, only *M. splendida* (accession no. MZ295219) tested positive for phytoplasma strains associated with the bamboo species, which is the first record in the world [27]. Nevertheless, leafhoppers may play a crucial role in the transmission of bamboo phytoplasma strains in the future as well as many other monocot and dicot hosts [28]; theseneed further studies as the natural reservoirs in epidemiology of phytoplasmas associated with the bamboo species. The taxonomic identity of leafhopper-associated phytoplasmal diseases with the bamboo species, weeds and food crops should be more accurately determined and classified by employing various morpho-molecular tools. The variability of colouration in *M. splendida* has led to confusion in its taxonomy; for this reason, the main aim of this paper wasto integrate the morphological and molecular data—along with the distribution, the biology and detailed illustrations—to pave the way for future research.

## 2. Materials and Methods

### 2.1. Explorations for the Collection of Insect Specimens

The terminology and the key for the identification used in this work followed Viraktamath and Webb [26]. The leafhopper specimens were collected from bamboo trees with the help of sweep nets during 2018–2022 from different localities in India. The collected specimens were separated based on their morphology and then kept in butter paper covers with detailed labels, including the name of the locality, date of collection, location coordinates, elevation, name of the collector and host plant. The butter paper covers were brought to the laboratory in specimen-storing boxes, which were sorted and processed for further studies.

### 2.2. Morphological Identification

Photographs were taken with a Leica DFC 425C digital camera attached to a Leica M205FA stereo zoom automontage microscope. Male genitalia dissections were performedbased on the procedures adopted by Oman and Knight [29,30]. The material examined wasdeposited in the National Pusa Collection, Division of Entomology, ICAR-Indian Agricultural Research Institute, New Delhi, India (NPC).

### 2.3. Molecular Identification and Analysis

In leafhopper taxonomy, males are more important for the species identification. For the mitochondrial cytochrome oxidase subunit I (mtCOI) analysis, the DNA was extracted from a single male specimen of each morphological type, according to the manufacturer’s protocol for the QIAGEN QIAamp DNA Investigator Kit. The number of Type 3 and Type 10 male specimens were few; hence, they were not used for the DNA extraction. The isolated DNA was stored at −20 °C until required. The barcode region of mtCOI was amplified using the primers LCO1490: 5’-GGT- CAACAAATCATAAAGATATTGG-3’ and HCO2198:5’-TAAACTTCAGGGTGAC- CAAAAAATCA-3’. The PCR was performed with a total reaction volume of 25 μL using DNA polymerase (FermentasGmBH, St. Leon-Rot, Germany) under the following cycling protocol: 4 min hot start at 94 °C; 35 cycles of denaturation for 30 s at 94°; annealing for 60 s at 47 °C; elongation for 50 s at 72 °C; and a final extension at 72 °C for 8 min in a C1000 thermal cycler. The reaction mixture included (as described by the KOD FXpuregene TM manufacturer’s protocol) 4 μL of the DNA template, 12.5 μLof2× PCR buffer, 10 μL of 2 mM dNTP and 1 unit of the TAQ (KODFX) enzyme. The forward and reverse primers were 0.3 μM each at the final concentration. The products were checked on 0.8% agarose gel and visualised under UV using Alphaview^®^software, version1.2.0.1. The amplified products were sequenced at SciGenomePvt Ltd. (Cochin, India). The sequences were assembled and aligned with BioEdit(version 7.0.0) and deposited in the NCBI GenBank. Accession numbers were obtained (Table 1). To determine the sequence divergence between the morpho variants of *M. splendida*, we performed a Basic Local Alignment Search Tool (BLAST) was usedto query the National Center for Biotechnology Information (NCBI) nucleotide database for COI sequence data inblast searches. The sequences were aligned with the help of CLUSTAL W and Pairwise-corrected (K2P) genetic distancewas prepared with MEGA version 10 using Kimura-2-Parameterdistance model (Table 2).

## 3. Results

### 3.1. Taxonomic Status

#### 3.1.1. *Mukaria splendida* Distant, 1908

Hemiptera: Auchenorrhyncha: Cicadomorpha: Membracoidea: Cicadellidae: Deltocephalinae: Mukariini.

#### 3.1.2. *Mukaria splendida* Distant, 1908: 270–271

Description of species: Glossy or shining black to straw or yellowcolour, small to medium-sized and depressed leafhoppers. An examination of all specimens of *Mukaria splendida* Distant from different localities indicated the presence of 10 colour morphs from each sex, based on variations in the black and yellow pattern on the pronotum and forewings (see Table 3 and Table 4 and Figure 1, Figure 2 and Figure 3).

**Male Genitalia:** The pygofer was 2.5× wider than long, with a prominent T-shaped caudal ventral process on each side and an anal tube with microsetae (Figure 4a,b). Subgenital plates were wider, extending beyond the pygofer both laterally and posteriorly with a few macrosetae (Figure 4c). The valve triangle (Figure 4c) was articulated with pygofer. The style was broad at the base and slightly curved in the posterior region;the apophysis was slenderer(Figure 4d). The connective was equal in length. The aedeagus was broad and the aedeagal shaft was bifurcate; each shaft had adorsolateral plate-like process near the apical gonopore [26] (Figure 4e–g).

**Female genitalia:** The sternite VII was about three times as wide as long medially, the posterior margin was concave with a median broad convex lobe and the lateral angles were acutely rounded. There was a pygofer spinose in the posterior half; the ovipositor id not exceed the pygofer. Valvula I and valvula II were almost straight, with a prominent tooth; these were well-separated from each other [26] (Figure 4h–m).

**Material examined**: Total: 161♂, 186♀; India.

Colour morph1: Total: 47♂, 23♀; India: 2♂, Andhra Pradesh: Jytothikshetram (14°49’71” N 78°91’47” E) 111 m, 18.i.2020, net sweep, Coll. Ramaiah M; Tirupati (13°62’24” N 79°42’85” E) 147 m, 20.i.2020, net sweep, Coll. Ramaiah M; 3♂, 2♀,Telangana: Hyderabad (17°19’17.44” N 78°28’35.32” E) 542.3 m, 27.i.2020, net sweep, Coll. Ramaiah M; 2♂, Assam: Jakhalbandha (26°57’08” N 93°00’26” E) 76.89 m, 9.iii.2020, net sweep, Coll. Ramaiah M; 8♂, 3♀ Nagaland: Medziphema (26°72’48” N 94°19’56” E) 295 m, 16.iii.2020, net sweep, Coll. Ramaiah M; 11♂, 6♀, New Delhi: Pusa campus, IARI (28.080° N 77.120° E) 228.61 m, 5.xii.2019,net sweep, Coll. Ramaiah M; 2♂, Rajasthan: Ajmer (26°26’59.14” N 74°38’28.18” E) 480 m, 01.iii.2022, net sweep, Coll. Stuti; 9♂, 12♀, Uttarakhand: FRI, Dehradun (30.3455° N 78.0132° E) 430 m, 24.xi.2020, net sweep, Coll. Ramaiah M; 5♂, Maharashtra: Chikhaldara forest (21°24’25.3” N 77°21’27.2” E), 19.xii.2017, net sweep, Coll. Akash; 5♂, Himachal Pradesh: Berthin (Bilaspur), (31°19’30.96” N 76°38’21.42” E) 673 m, 08.x.2018, Coll. Akash.

Colour morph2: Total: 19♂, 55♀;India:5♀, Andhra Pradesh: Tirupati (13°62’24” N 79°42′85” E) 147 m, 20.i.2020, net sweep, Coll. Ramaiah M; 3♂, Himachal Pradesh: Solan, (Nauni) (30°51’44.7444” N77°10’9.1488” E) 1275 m, 25.iii.2022, Coll. Ramaiah M; 5♀, Telangana: Hyderabad (17°19’17.44” N 78°28’35.32” E) 542.3 m, 27.i.2020, net sweep, Coll. Ramaiah M; 5♀,Assam: Jakhalbandha (26°57’08” N 93°00’26” E) 76.89 m, 9.iii.2020, net sweep, Coll. Ramaiah M; 3♂, 6♀, Nagaland: Medziphema (26°72’48” N 94°19’56” E) 295 m, 16.iii.2020, net sweep, Coll. Ramaiah M; 7♂, 13♀, New Delhi: Pusa campus, IARI (28.080° N 77.120° E) 228.61 m, 5.xii.2019, net sweep, Coll. Ramaiah M; 2♂, 6♀, Rajasthan: Ajmer (26°26’59.14” N 74°38’28.18” E) 480 m, 01.iii.2022, net sweep, Coll. Stuti; 4♂, 20♀, Uttarakhand: FRI, Dehradun (30.3455° N 78.0132° E) 430 m, 24.xi.2020, net sweep, Coll. Ramaiah M.

Colour morph3: Total: 24♂, 19♀; India: 3♀, Telangana: Hyderabad (17°19’17.44” N 78°28′35.32” E) 542.3 m, 27.i.2020, net sweep, Coll. Ramaiah M;5♂, 3♀, Nagaland: Medziphema (26°72’48” N 94°19’56” E) 295 m, 16.iii.2020, net sweep, Coll. Ramaiah M; 4♂, 7♀, New Delhi: Pusa campus, IARI (28.080° N 77.120° E) 228.61 m, 5.xii.2019,net sweep, Coll. Ramaiah M; 2♂, Rajasthan: Ajmer (26°26’59.14” N 74°38’28.18” E) 480 m, 01.iii.2022, net sweep, Coll. Stuti; 5♂, 6♀, Uttarakhand: FRI, Dehradun (30.3455° N 78.0132° E) 430 m, 24.xi.2020, net sweep, Coll. Ramaiah M.

Colour morph4: Total: 17♂, 23♀; India: 3♂, Telangana: Hyderabad (17°19’17.44” N 78°28’35.32” E) 542.3 m, 27.i.2020, net sweep, Coll. Ramaiah M; 2♂, Assam: Jorhat (26°72‘48” N 94°19‘56” E) 116 m, 2.iii.2020, net sweep, Coll. Ramaiah M; 3♂, Nagaland: Medziphema (26°72’48” N 94°19’56” E) 295 m, 16.iii.2020, net sweep, Coll. Ramaiah M; 4♂, 6♀, New Delhi: Pusa campus, IARI (28.080° N 77.120° E) 228.61 m, 5.xii.2019,net sweep, Coll. Ramaiah M; 2♀, Rajasthan: Ajmer (26°26’59.14” N 74°38’28.18” E) 480 m, 01.iii.2022, net sweep, Coll. Stuti;3♂, 6♀,Uttarakhand: GBPAUT, Pantnagar (29°02’60.00” N 79°30’59.99” E) 243.84 m, 26.xi.2020, net sweep, Coll. Ramaiah M; 2♂, 5♀, Maharashtra: Chikhaldara forest (21°24’25.3” N 77°21’27.2” E), 19.xii.2017, net sweep, Coll. Akash; 4♀, Himachal Pradesh: Berthin (Bilaspur) (31°19’30.96” N 76°38’21.42” E) 673 m, 08.x.2018, Coll. Akash.

Colour morph5: Total: 19♂, 17♀; India: 7♂, Andhra Pradesh; 2♂, Telangana; 2♀, Assam; 2♂, 3♀, Nagaland; 3♂, 5♀, New Delhi; 1♂, 2♀, Rajasthan; 4♂, 2♀,Uttarakhand; 3♀, Himachal Pradesh.

Colour morph6: Total: 4♂, 11♀; India: 2♂, 5♀, New Delhi; 2♂, Uttarakhand;6♀, Maharashtra.

Colour morph7: Total: 11♂, 13♀;India: 7♀, Andhra Pradesh; 1♂, Nagaland; 4♂, 4♀, New Delhi; 2♀, Rajasthan; 4♂, Uttarakhand; 2♂, Himachal Pradesh.

Colour morph8: Total: 6♂, 8♀; India: 2♂, 8♀, New Delhi; 2♂, Uttarakhand; 2♂, Himachal Pradesh.

Colour morph9: Total: 8♂, 10♀; India: 1♀, Telangana; 1♂, 2♀, Assam; 1 ♀, 3♂, Madhya Pradesh: Hoshangabad, (21°48’53” N 76°21’55” E) 278 m, 26.iii.2022, Coll. Anand; 2♀, Nagaland; 4♂, 2♀, New Delhi; 3♀, Uttarakhand.

Colour morph10: Total: 4♂, 6♀; India: 1♂, Assam; 1 ♀, 4♂, Madhya Pradesh: Hoshangabad (21°48’53” N 76°21’55” E) 278 m, 26.iii.2022, Coll. Anand;2♀, Nagaland; 2♂, 3♀, New Delhi.

**Distribution:** Bangladesh. India: Andhra Pradesh, Assam, Bihar, Gujarat, Himachal Pradesh, Karnataka, Kerala, Madhya Pradesh*, Maharashtra*, Nagaland*, New Delhi*, Punjab, Rajasthan*, Tamil Nadu, Telangana*, Uttarakhand* and West Bengal. Pakistan: Sindh (* new locality records).

Host plant: Bamboo.

**Biology:** Feeds on bamboo; eggs are deposited after piercing the leaf tissue (Figure 5a,b). The nymphs are white to pale yellow with numerous black spots on the body (Figure 5a). The adults are shiny black with a yellow or white pattern on the pronotum and forewings (Figure 5b).

### 3.2. Molecular Identification

The diagnosis of any insect pest species is very important for the success of any pest management program. In this paper, integrative taxonomy based on a morphomolecular analysis was used to confirm the species identity. The DNA sequences generated (Table 1) matched 97.54–100% ofthe available relative sequences. The molecular marker mtCOI gene, a reputed marker for species identification, was used in the molecular analysis [31].

## 4. Discussion

Colour polymorphism is a fascinating facet of many organisms that maintain unique variations for growth and development. The coexistence of two or more-colourmorphs have been reported in mantoids, cicadids, damselflies and lepidopterans [32,33,34]. Similarly, the existence of colour polymorphism has beenwidely recorded in Orthopterans through crypsis, background matching and homochromy [32] for protection against natural enemies and survival. In the present study, the colour polymorphism of *M. splendida* was highlighted, along with additional geographical distribution records.The integration of the morpho-molecular data confirmed that all the colour morphs belonged to the same species, *M. splendida*. The molecular marker mtCOI gene, a reputed marker for species identification [32], was used.

Ten different colour morphs in *M. splendida* could be placed into ten types based on the colour pattern on the pronotum and forewings (see Table 3 and Table 4 and Figure 1, Figure 2 and Figure 3). Type1 in males and Type2 in females werethe most prominent of the morphs compared with the other types (Table 3 and Table 4). Morphologically, all specimens differed in their habitus colouration whereas the genitalia of all specimens, irrespective of colour, were similar. The adults of *M. splendida* are morphologically very similar to other species of *Mukaria*, but can be readily differentiated by theirpeculiar processes of the aedeagus and the T-shaped ventral pygofer process. In addition, a greater acute posterior angle of the male valve and the sinuate posterior margin of the female sternite VII were present. *M. splendida* is widely distributed throughout India [20]. Recently, it was recorded as a vector for the *Candidatus Phytoplasma australasia* (16SrII-D) phytoplasma subgroup, which is associated with the bamboo species; it is major threat to the bamboo ecosystem [27] and is one of the major insect pests of bamboo [13].

The colouration in lepidopteran larvae is reported to be highly dynamic. In *Helicoverpa armigera*, larval colour variation was recorded ranging from velvety black to yellowish green [35,36]. Similarly, other larval colours—*viz*., green, fawn, pink, yellow or brown and very dark to light green or pink—were also [37,38]. Additionally, so many workers reported instar-wise colour variations [35,39,40,41,42,43]. Suvarna Patil (2005) observed about six to seven colour morphs of *H. armigera* in differentsurveyed hosts [43]. Yamasaki et al. (2009) reported that larval colouration in *H. armigera* Hubner was primarily determined by the host plant and the plant part on which the larvae fed [44]. Sampath et al. (2017) recorded six colour morphs thatshowed a significant positive correlation with a minimum temperature and rainfall in *Maruca vitrata* [45]. Apart from a high temperature and humidity, a high food moisture content and a low individual density are also known to drive green body colouration in grasshoppers [32,33]. These clearly revealed that the colour variation is also attributed to the nutritional status of the associated host as well as genetic factors and the location. The levels of pigments in insect diets also play an important role in the insect colouration. According to Ramos and MoralloRajesus (1976), beta-carotene in anartificial diet influenceddifferences in the larval colour forms of *Maruca vitrata* [46]. Maragal (1990) noted green-coloured morphs when *Helicoverpa* larvae were fed on flower petals rich in a yellow colour [47].

In Hemiptera, especially in thecicadellidae group, fewerstudies have been undertakenon the existence of colour polymorphism. Recently, Li et al. (2012) recognized eight colour forms in the leafhopper species *Macropsis notata* (Prohaska) from China [48]. Likewise, Stuti and Meshram (2019) recorded a distinct morph in *Anagonalia lapnanensis* [49]. Similarly, Praveen and Sabita (2019) also recorded four morphs among populations of *M. splendida* Distant collected from different locations [20]. In our study, location-specific variations were not observed as almost all ofthe ten morphs of each sex could be collected from the same location. Viraktamath and Webb (2019) mentioned in their monographthat their *M. splendida* individuals were glossy black and recorded only one each for males (our Type4) and females (our Type8) as represented [26] (Table 3 and Table 4). To establish the colour morphs as a single species, a molecular analysis was undertaken. The pairwise distance matrix (Table 2) showed a genetic distance ranging from 0.00–0.05, which wasnegligible according to the divergence rule [50]. Thus, it wasinferred that the differencesfound in the colour morph specimens weredeemed to be variations in *M. splendida* rather than being froma different species. Morphologically, all specimens of *Mukaria splendida* from different localities showeda different habitus colouration whereas the genitalia of all specimens, irrespective of colour, weresimilar. This colour polymorphism may be due to the effect of its association with different bamboo species and varieties. In order to understand the effect of the ecosystem on the intraspecific variations in *M. splendida*, different molecular markers must be included. This study helps tocorrectly identify *M. splendida*,which in turn helps with pest management.

## 5. Conclusions

An authentic species-level diagnosis is often a key to successful insect pest management and bio-security. The results of the present study provide information about the morph-molecular confirmation of colour polymorphism as well as additional geographical distribution records and biology. This group exclusively feeds on and causes damage to bamboo and is reported to beone of the major insect pests on bamboo. Thus, thecareful monitoring and timely reporting of the pest status are necessary to avoid any future outbreaks as this species is reported to be a vector for the *Candidatus Phytoplasma australasia* (16SrII-D) phytoplasma subgroup. The integration of the morphology with a molecular analysis will aid the authentic identification of *M. splendida*. The colour polymorphism may be of an evolutionary significance.

## Figures and Tables

**Figure 1 insects-14-00044-f001:**
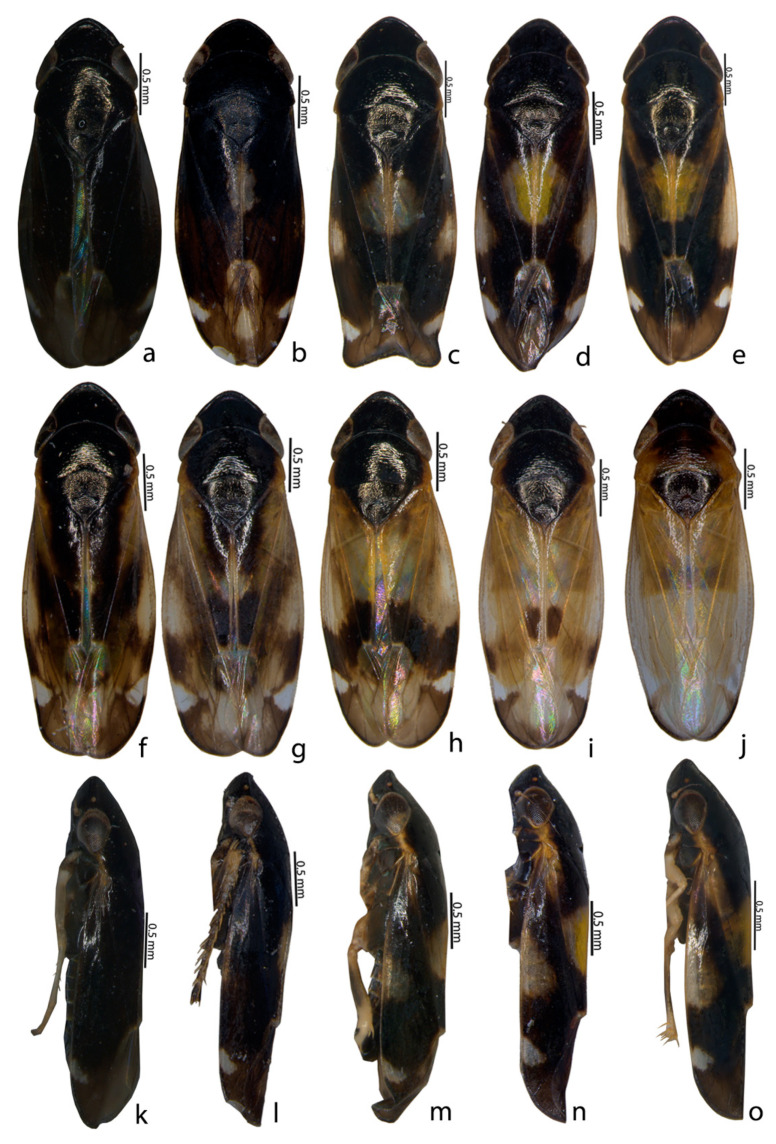
Colour polymorphism in *Mukaria splendida* Distant. (**a**–**j**) ♂ dorsal habitus; (**k**–**o**) ♂ lateral habitus.

**Figure 2 insects-14-00044-f002:**
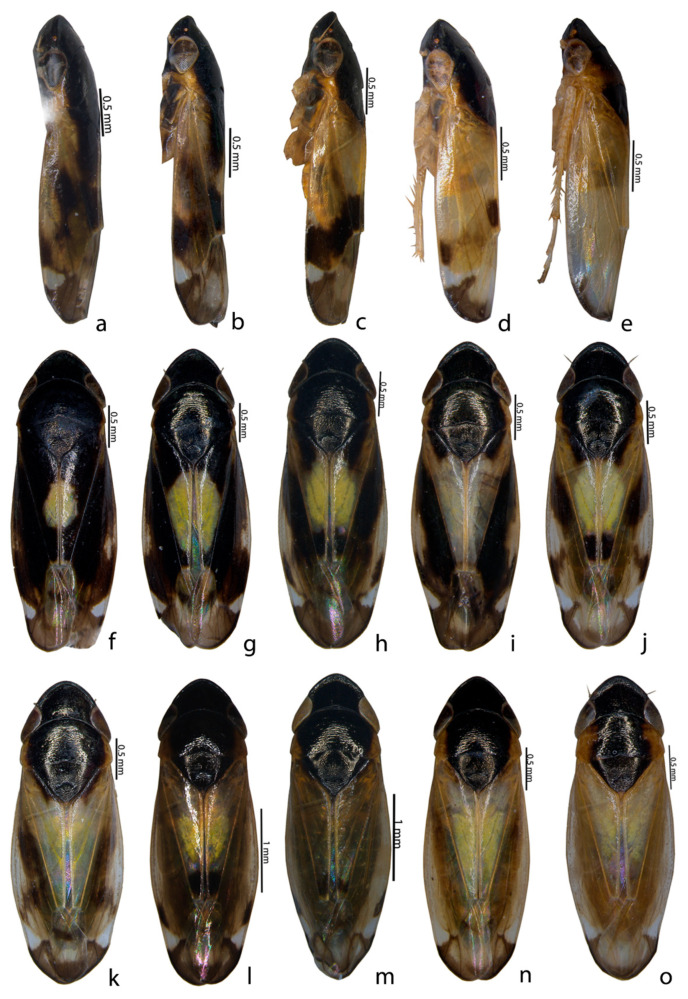
Colour polymorphism in *Mukaria splendida* Distant. (**a**–**e**) ♂ lateral habitus; (**f**–**o**) ♀ dorsal habitus.

**Figure 3 insects-14-00044-f003:**
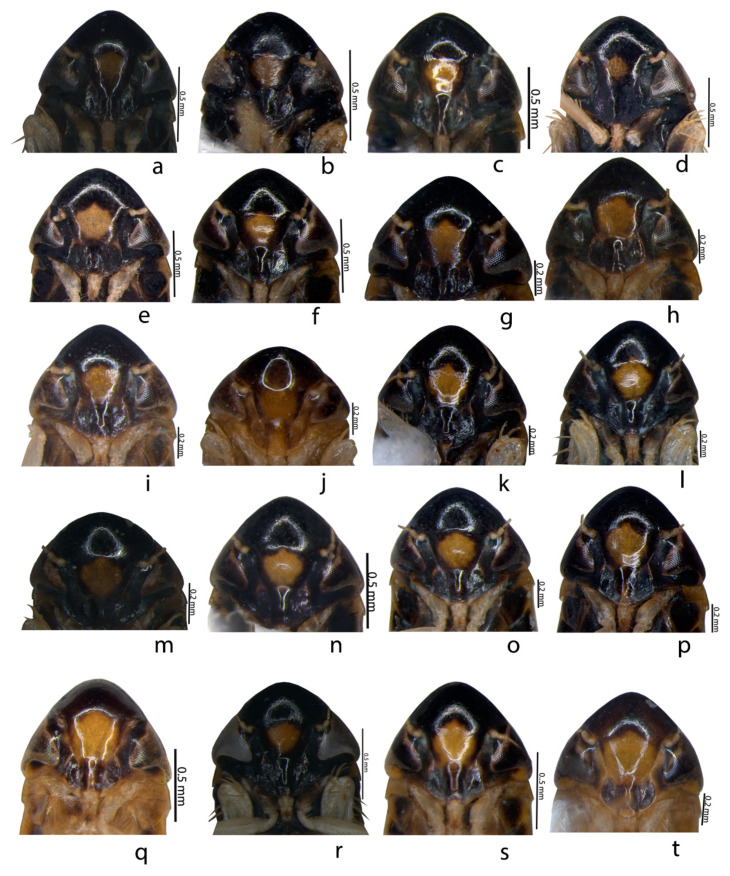
Colour polymorphism in *Mukaria splendida* Distant. (**a**–**j**) ♂ face; (**k**–**t**) ♀ face.

**Figure 4 insects-14-00044-f004:**
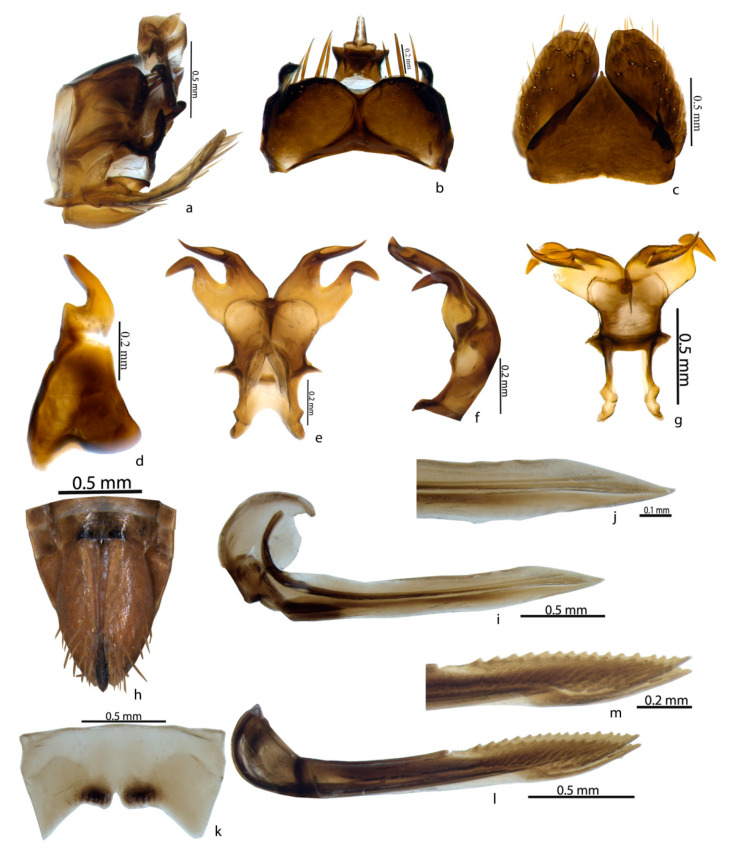
♂ and ♀ genitalia of *Mukaria splendida* Distant. *Male:* (**a**)—genital capsule, lateral view; (**b**)—pygofer, dorsal view; (**c**)—valve and subgenital plate, dorsal view; (**d**)—style, dorsal view; (**e**)—connective and aedeagus, anterodorsal view; (**f**)—aedeagus, lateral view; (**g**)—connective and aedeagus, ventral view. *Female:* (**h**)—female ovipositor; (**i**)—valvula I; (**j**)—valvula I, apex magnified; (**k**)—sternite VII; (**l**)—valvula II; (**m**)—valvula II, apex magnified.

**Figure 5 insects-14-00044-f005:**
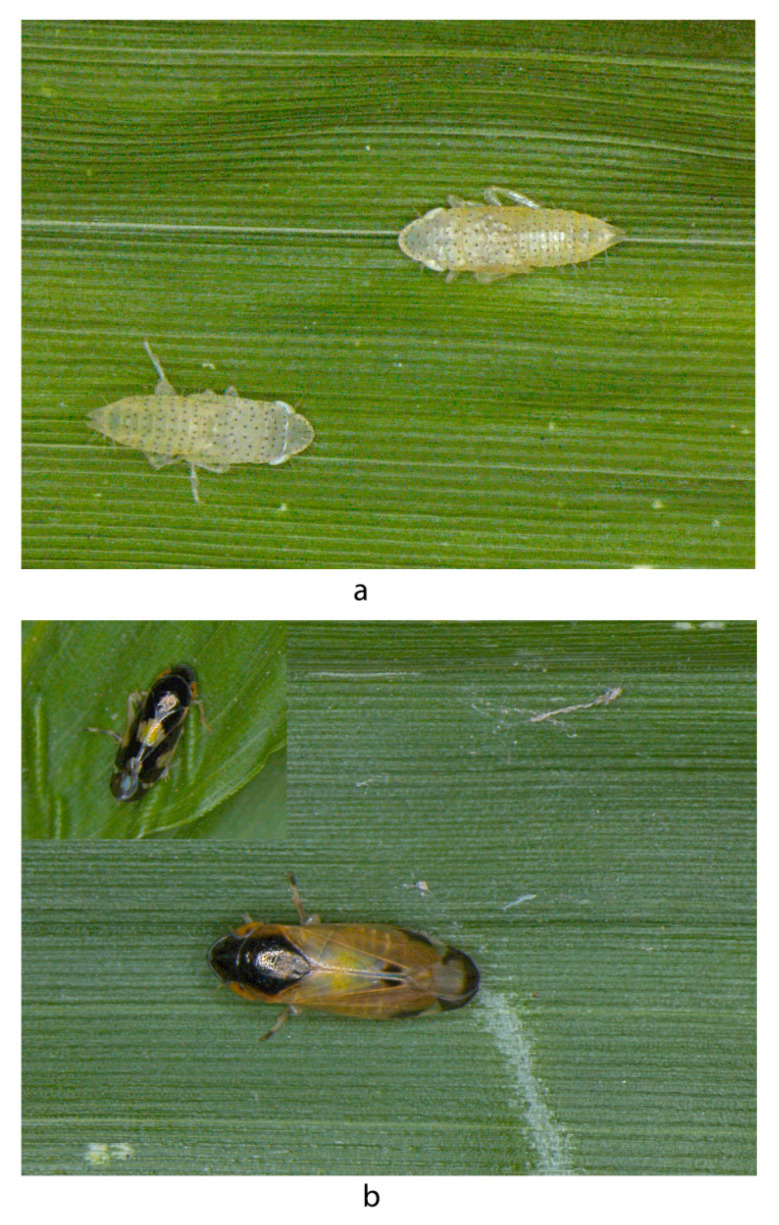
Nymphs and adult *Mukaria splendida* Distant. (**a**)—nymph; (**b**)—adult.

**Table 1 insects-14-00044-t001:** Details of the sample data used in the molecular analysis of the *M. splendida* (male) population.

Sl. No.	Tribe	Colour Morph	% Similarity	GenBank Sequence ID
1	Mukariini	Type 1	100	OM877494
2		Type 2	97.54	OP622876
3		Type 4	99.54	OP616038
4		Type 5	99.38	OM869458
5		Type 6	100	OM345004
6		Type 7	98.93	OP617462
7		Type 8	100	OM868261
8		Type 9	99.61	OP622873

Note: *M. splendida*. Sequences were submitted to NCBI from the present study.

**Table 2 insects-14-00044-t002:** Pairwise-corrected (K2P) genetic distance among the colour morphs of the *M. splendida* species.

Colour Morph ♂	T1	T2	T4	T5	T6	T7	T8	T9
Type 1: *M. splendida*		0.010	0.001	0.000	0.001	0.003	0.000	0.002
Type 2: *M. splendida*	0.056		0.015	0.015	0.015	0.014	0.006	0.008
Type 4: *M. splendida*	0.001	0.125		0.002	0.003	0.004	0.002	0.002
Type 5: *M. splendida*	0.000	0.123	0.003		0.003	0.002	0.000	0.001
Type 6: *M. splendida*	0.001	0.125	0.006	0.009		0.004	0.002	0.004
Type 7: *M. splendida*	0.005	0.118	0.010	0.004	0.014		0.003	0.003
Type 8: *M. splendida*	0.000	0.016	0.002	0.000	0.002	0.006		0.002
Type 9: *M. splendida*	0.002	0.032	0.003	0.001	0.011	0.007	0.002	

**Table 3 insects-14-00044-t003:** Colour pattern and body length of males of *M. splendida*.

Type	Colouration Pattern	Body Length (Including Wings)
1	Body: black; forewing: no spot on midlength, spot on costal region near outer apical cell; forewing apex: black; pronotum: black (Figure 1a,k and Figure 3a).	3.11 mm
2	Body: black; forewing: one linear spot at midlength, spot on costal region near outer apical cell; forewing apex: black; pronotum: black (Figure 1b,l and Figure 3b).	3.31 mm
3	Body:black;forewing: one circular white spot at midlength, oblique spot on costa, subtriangular spot near outer apical cell; forewing apex:black;pronotum: black (Figure 1c,m and Figure 3c).	3.01 mm
4	Body: black; forewing: yellow spot enlarged at midlength, oblique spot-on costa, subtriangular spot near outer apical cell; forewing apex: black; pronotum: black except lateral margin (Figure 1d,n and Figure 3d).	3.36 mm
5	Body: black;forewing:yellow spot at midlength extending to lateral margin ofpronotum, oblique spot on costa, subtriangular spot near outer apical cell; forewing apex: black; pronotum: black except lateral margin (Figure 1e,o and Figure 3e).	3.22 mm
6	Body: dark brown;forewing: black, spot on costal region extended to pronotum, subtriangular spot near outer apical cell; forewing apex: pale; pronotum: black except lateral margin (Figure 1f, Figure 2a and Figure 3f)	3.42 mm
7	Body: dark brown; pattern as in No. 6 except spot at midlength (Figure 1g, Figure 2b and Figure 3g).	3.37 mm
8	Body: dark brown; forewing: anterior half hyaline below which was thick black W-shaped marking with subtriangular spot-on lateral margin; pronotum: black except lateral margin (Figure 1h, Figure 2c and Figure 3h)	3.28 mm
9	Body: brown; forewing: hyaline except brown spot on posterocostal margin, followed by subtriangular spot, clavus distal with two longitudinal black spots; apical margin: black; pronotum: black except lateral margin (Figure 1i, Figure 2d and Figure 3i)	3.02 mm
10	Body: brown; forewing: hyaline with black bordered apical margin; pronotum: black except lateral margin (Figure 1j, Figure 2e and Figure 3j)	3.02 mm

**Table 4 insects-14-00044-t004:** Colour pattern and body length of females of *M. splendida*.

Type	Colouration Pattern	Body Length (Including Wings)
1	Body: black; forewing: yellow spot at midlength, oblique spot-on costa, subtriangular spot near outer apical cell; forewing apex: pale; pronotum: black (Figure 2f and Figure 3k)	3.66 mm
2	Body: black; forewing: clavus yellow except at both margins, oblique spot-on costa, subtriangular spot near outer apical cell; forewing apex: pale; pronotum: black (Figure 2g and Figure 3l)	3.12 mm
3	Body: black; forewing: clavus yellow extending to pronotum except at both margins, oblique spot-on costal region extending to subtriangular spot near outer apical cell; forewing apex: black; pronotum: black except lateral margin (Figure 2h and Figure 3m).	3.38 mm
4	Body: black; forewing: white clavus extended to pronotumexcept at distal margin, oblique spot-on costa, subtriangular spot near outer apical cell; forewing apex: black; pronotum: black except lateral margin (Figure 2i and Figure 3n).	3.56 mm
5	Pattern as in No. 4 except black margins on clavus; forewing apex: pale (Figure 2j and Figure 3o).	3.24 mm
6	Pattern as in No. 4 except paler colour and clavus proximal margin with black line (Figure 2k and Figure 3p).	3.53 mm
7	Body: brown; forewing: clavus distal with two longitudinal black spots, oblique spot-on costa, subtriangular spot near outer apical cell; forewing apex: black; pronotum: black except lateral margin (Figure 2l and Figure 3q).	3.69 mm
8	Body: brown; pattern as in No. 7 except hyaline wings with dark brown body;forewing apex: pale (Figure 2m and Figure 3r).	3.56 mm
9	Body: brown; pronotum: black except ½ of portion (Figure 2n and Figure 3s)	3.59 mm
10	Body: brown; pattern as in No. 9 except forewing apex: pale; pronotum: black except ¼ portion (Figure 2o and Figure 3t).	3.45 mm

## Data Availability

All specimens are deposited in National Pusa Collections, New Delhi stated in the paper. All DNA sequences data has been deposited in GenBank (accession number OM877494,OP622876, OP616038, OM869458, OM345004, OP617462, OM868261, OP622873).

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
