# Peer review of "Integrative Approaches Establish Colour Polymorphism in the Bamboo-Feeding Leafhopper Mukaria splendida Distant (Hemiptera: Cicadellidae) from India"

_insects, 2023, doi:10.3390/insects14010044_

Round 1

Reviewer 1 Report

This manuscript documents color variation in the leafhopper Mukaria splendida Distant.  It is illustrated very nicely, showing the different forms.

Some criticisms:

The slight differences between some of the forms suggests that the variation is on a continuous spectrum rather than as discrete forms as the authors seem to suggest.  This topic- continuous vs. discrete variation should be discussed.  And if they are discrete forms, the authors need to state how that was determined, e.g. how many specimens of each discrete form were observed? 

Table 1 has a column for % identity but it is not stated what the listed sequence is being compared to, making this table problematic, and affects how the rest of the data is interpreted. 

In the Discussion, the results in Table 2 are discussed but the numbers used differ from the table 2. Based on Table 2, the range of pairwise distances is 0.0% to 0.125%, but a range of 0.0% to 0.05% is stated in Discussion.  How can such an error be made?

A much more in-depth discussion needs to be made by the authors on the criteria for determining intra-species variation vs. inter-species variation.  The "2% rule" is sometimes generally used, but there in controversy over this.  What has been found and used in other species of Cicadellidae?  Since other species of Mukaria were not sequenced in this study, what are the limitations on what can be determined with the data that is presented?

Also, please clarify that the values in Table 2 represent %, and not raw corrected pairwise distances as is often reported in other similar studies.

Author Response

Dear sir,

The reason behind pairwise distance matrix results are not uniform in the length of the Sequence (bp), because of that this minor variation came 

all corrections suggested by you included in the revised manuscript 

Reviewer 2 Report

GENERAL COMMENTS

1. “Simple summary” is lacking (it should be added to the MS).

2. Please provide the Latin name of bamboo when for the first time mentioned in the text.

3. The citation system in the MS is entirely erroneous. It should be corrected throughout the manuscript. According to the “Instructions for Authors”, the references should be cited in numerical order, and the reference numbers should be placed in square brackets [ ], and placed before the punctuation.

4. The English language needs corrections by a native English speaker. I have corrected only some language mistakes (directly in the text).

5. Male and female graphic symbols provided in the material examined sections, and the figures' captions, should be improved (there are too “massive”).

SPECIFIC COMMENTS

1. Introduction

Please provide a short summary regarding the molecular studies conducted till now on Mukaria splendida.

 2. Material and methods

2.3.Molecular analyses

How many male and female specimens have been used for DNA extraction and sequencing must be clearly marked. In such studies, it is essential to include as many sequences of the studied morphs as possible in the analyses.

3. Results

Material examined section: p. 9, line 21: (*New locality records). I suggest changing this fragment to (*New state records) or something similar. All “localities” provided in the paper are new.

Molecular identification section: p. 10, lines 220-225. This section is poorly prepared. Please improve its contents.

Other suggestions were put directly into the text.

Author Response

Dear sir,

All corrections suggested by you included in the revised manuscript 

Thank you very much sir

Round 2

Reviewer 2 Report

Still, there are several points, which should be corrected in the MS:

1.     “Simple summary” is still lacking (it should be added to the MS).

2.     The citation system in the MS was not corrected and is still entirely erroneous. It should be corrected throughout the manuscript. According to the “Instructions for Authors”, the references should be cited in numerical order, and the reference numbers should be placed in square brackets [ ], and placed before the punctuation.

3.     The English language still needs some corrections.

Author Response

Dear sir

PFA regarding review report (reply to reviewer 2) 
